# Metabolomics and Transcriptomics Reveal the Response Mechanisms of *Mikania micrantha* to *Puccinia spegazzinii* Infection

**DOI:** 10.3390/microorganisms11030678

**Published:** 2023-03-07

**Authors:** Xinghai Ren, Guangzhong Zhang, Mengjiao Jin, Fanghao Wan, Michael D. Day, Wanqiang Qian, Bo Liu

**Affiliations:** 1Shenzhen Branch, Guangdong Laboratory of Lingnan Modern Agriculture, Genome Analysis Laboratory of the Ministry of Agriculture and Rural Affairs, Agricultural Genomics Institute at Shenzhen, Chinese Academy of Agricultural Sciences, Shenzhen 518000, China; 2Key Laboratory of Entomology and Pest Control Engineering, College of Plant Protection, Southwest University, Chongqing 400715, China; 3Department of Agriculture and Fisheries, Ecosciences Precinct, GPO Box 267, Brisbane, QLD 4001, Australia

**Keywords:** transcriptome, metabolome, immune response, biological control, Pucciniales

## Abstract

*Mikania micrantha* is one of the worst invasive species globally and can cause significant negative impacts on agricultural and forestry economics, particularly in Asia and the Pacific region. The rust *Puccinia spegazzinii* has been used successfully as a biological control agent in several countries to help manage *M. micrantha*. However, the response mechanisms of *M. micrantha* to *P. spegazzinii* infection have never been studied. To investigate the response of *M. micrantha* to infection by *P. spegazzinii*, an integrated analysis of metabolomics and transcriptomics was performed. The levels of 74 metabolites, including organic acids, amino acids, and secondary metabolites in *M. micrantha* infected with *P. spegazzinii*, were significantly different compared to those in plants that were not infected. After *P. spegazzinii* infection, the expression of the TCA cycle gene was significantly induced to participate in energy biosynthesis and produce more ATP. The content of most amino acids, such as L-isoleucine, L-tryptophan and L-citrulline, increased. In addition, phytoalexins, such as maackiain, nobiletin, vasicin, arachidonic acid, and JA-Ile, accumulated in *M. micrantha*. A total of 4978 differentially expressed genes were identified in *M. micrantha* infected by *P. spegazzinii*. Many key genes of *M. micrantha* in the PTI (pattern-triggered immunity) and ETI (effector-triggered immunity) pathways showed significantly higher expression under *P. spegazzinii* infection. Through these reactions, *M. micrantha* is able to resist the infection of *P. spegazzinii* and maintain its growth. These results are helpful for us to understand the changes in metabolites and gene expression in *M. micrantha* after being infected by *P. spegazzinii*. Our results can provide a theoretical basis for weakening the defense response of *M. micrantha* to *P. spegazzinii*, and for *P. spegazzinii* as a long-term biological control agent of *M. micrantha*.

## 1. Introduction

*Mikania micrantha* Kunth (Asteraceae), commonly called ‘mile-a-minute’, is a rapidly growing vine native to tropical America [1,2]. It is considered one of the world’s worst weeds, invading many countries in Asia and Oceania [1,2] and has been listed in China as a plant species with national quarantine concerns by the Forestry Administration [3,4]. *Mikania micrantha* has been reported in seven provinces in southern China, with the main infestations in Guangdong, Guangxi, and Yunnan provinces [5]. It affects biodiversity and a wide range of agriculture and forestry enterprises, reducing productivity through competition [1]. *Mikania micrantha* is also known as a plant killer because its rapid growth enables it to completely smother crops or trees and block sunlight, preventing flowering and fruiting [1,3].

The rust fungus, *Puccinia spegazzinii* de Toni (Uredinales: Pucciniaceae), has been introduced into several countries, including China, to help control *M. micrantha* [1]. *Puccinia spegazzinii* is a microcyclic and autoecious rust with a life cycle of 19–21 days [6]. Teliospores and basidiospores of *P. spegazzinii* have only been recorded in the field [7]. *Puccinia spegazzinii* is known to attack only *M. micrantha* and to a much lesser extent, *Mikania cordata* (Burm. F.) B. L. Robinson. Its potential as a biological control agent is because it is particularly damaging to the leaves, stems and petioles of *M. micrantha*, reducing growth rates and flowering, and can tolerate a wide range of environmental conditions in which *M. micrantha* grows [8,9,10,11].

Rust fungi (Pucciniales) are one of the largest groups of plant pathogens and the most damaging to plants in the world [12,13]. Infected plants can produce a range of defense responses to a pathogen [14]. The amount of adenosine triphosphate (ATP), which plays an important role in metabolite biosynthesis, signal transduction, and material transport, can be increased. These secondary metabolites play vital roles in performing or having passive physical and chemical barriers against pathogens. For instance, polyamines could change the mesophyll cell pre-penetration and penetration resistance mechanisms, e.g., as seen in oats in response to infection by *Puccinia coronata* f. sp. *avenae* W.P. Fraser and Ledingham (Pucciniaceae) [15].

Other responses relate to the phenylpropanoid, flavonoid and isoflavonoid metabolic pathway genes, which are involved in the production of phytoalexins as seen in *Medicago truncatula* Gaertn (Fabaceae) following infection by *Phakopsora pachyrhizi* Syd. and P. Syd. (Phakopsoraceae) [16]. In addition, the accumulation of phenolic, rutin, glucosinolate, flavonoid, fatty acid, and alkaloid phytoalexins in plants could promote the host defense against pathogens [17,18,19]. Defense-signalling networks, such as ethylene and the salicylic acid pathways, are also triggered when pathogens attempt to infect host plants [20], and several gene families, such as the MCM1, WRKY, MYB, and bZIP families, are involved in the response of plants to pathogens [21,22,23,24].

The metabolic responses in *M. micrantha* to infection by *P. spegazzinii* have never been studied. In this study, we have characterized *M. micrantha* as a model host pathosystem for *P. spegazzinii* to investigate the molecular mechanisms of the host response to infection. Using transcriptome and metabolome analyses, we show how levels of essential amino acids, ATP, and phytoalexins in *M. micrantha* change in response to infection by *P. spegazzinii*. In addition, we aim to show how genes related to biotic stress in *M. micrantha* are expressed following *P. spegazzinii* infection. This work will help understand the potential for the continuous use of *P. spegazzinii* in assisting in the management of *M. micrantha* in many countries.

## 2. Materials and Methods

### 2.1. Rust Inoculation and Sample Collection

*Puccinia spegazzinii* IMI 393075 was imported into China from Australia in December 2019, and a culture was maintained on potted *M. micrantha* plants at the Agricultural Genomics Institute at Shenzhen, Chinese Academy of Agricultural Sciences. *Mikania micrantha* plants were grown in a greenhouse with lighting (day: 16 h and night: 8 h) and a daily temperature range of 20–35 °C. For inoculation, plant tissue (leaves, stems, petioles) with mature *P. spegazzinii* pustules, indicated by a coppery-brown appearance, was suspended over healthy potted plants in a sealed plastic chamber maintained at 22 °C, 100% humidity, for 48 h of darkness (for details, see [7,11]). Then, 17 days following inoculation, infected leaves were collected in three biological replicates for the determination of metabolites. At this time, the teliospores had matured and become dark brown (Appendix A). After the teliospores mature, basidiospores can sprout from telia embedded in host tissues for the next infection cycle if the above conditions are met. As a control, three biological replicates of uninfected leaves from uninfected plants were also collected.

### 2.2. Metabolome Profiling

Following freezing by liquid nitrogen, the samples of infected leaves and uninfected leaves were ground into powder. A subsample of 100 mg powder from each of the three samples of infected and uninfected leaves was weighed, and the homogenate was resuspended with prechilled 500 μL 80% methanol and 0.1% formic acid by vortexing it well. The samples were incubated on ice for 5 min and were then centrifuged at 15,000× *g* at 4 °C for 10 min. The supernatant was diluted using LC-MS grade water, so the final concentration was 53% methanol. The samples were subsequently transferred to fresh Eppendorf tubes and were centrifuged at 15,000× *g* at 4 °C for 20 min. The supernatant was injected into the liquid chromatography–tandem mass spectrometry (LC-MS/MS) system analysis.

LC-MS/MS analyses were performed using an ExionLC AD system (SCIEX) coupled with a QTRAP 6500+ mass spectrometer (SCIEX) in Novogene Co., Ltd. (Beijing, China). The extracts were injected into a column (Xselect HSS T3, 2.5 μm, 2.1 × 150 mm, Waters) with a 20 min linear gradient at a 0.4 mL/min flow rate for the positive/negative polarity mode.

In the positive ion mode, samples were injected onto a BEH C8 Column (100 × 2.1 mm, 1.9 μm) using a 30-min linear gradient at a flow rate of 0.35 mL/min for the positive polarity mode. The eluents were eluent A (0.1% formic acid–water) and eluent B (0.1% formic acid–acetonitrile). The solvent gradient was set as follows: 5% B, 1 min; 5–100% B, 24.0 min; 100% B, 28.0 min; 100–5% B, 28.1 min; 5% B, 30 min. The QTRAP 6500+ mass spectrometer was operated in positive polarity mode with a curtain gas of 35 psi, collision gas set to medium, ion spray voltage of 5500 V, temperature of 500 °C, ion source gas of 1:55, and an ion source gas of 2:55.

In the negative ion mode, samples were injected onto an HSS T3 Column (100 mm × 2.1 mm) using a 25-min linear gradient at a flow rate of 0.35 mL/min for the negative polarity mode. The eluents were eluent A (6.5 mM ammonium bicarbonate–water) and eluent B (6.5 mM ammonium bicarbonate−95% methanol water). The solvent gradient was set as follows: 2% B, 1 min; 2–100% B, 18.0 min; 100% B, 22.0 min; 100–5% B, 22.1 min; 5% B, 25 min. A QTRAP 6500+ mass spectrometer was operated in positive polarity mode with a curtain gas of 35 psi, collision gas set to medium, ion spray voltage of −4500 V, temperature of 500 °C, ion source gas of 1:55, and an ion source gas of 2:55.

Based on the Novogene Database (novoDB, in-house database), multiple reaction monitoring (MRM) was used to detect the metabolites in each sample. The product ion (Q3) was used for metabolite quantification. The parent ion (Q1), Q3, retention time (RT), declustering potential (DP), and collision energy (CE) were used for metabolite identification. The data files generated by HPLC-MS/MS were processed using the SCIEX OS Version 1.4 to integrate and correct the peaks. The main parameters were set as follows: minimum peak height, 500; signal/noise ratio, 5; and Gaussian smooth width, 1. The area of each peak represents the relative content of the corresponding substance.

These metabolites were annotated using the Kyoto Encyclopedia of Genes and Genomes (KEGG) database (http://www.genome.jp/kegg/ (accessed on 10 October 2020)), Human Metabolome Database (HMDB) (http://www.hmdb.ca/ (accessed on 10 October 2020)) and Lipid Maps database (http://www.lipidmaps.org/ (accessed on 10 October 2020)). Principal components analysis (PCA) and partial least squares discriminant analysis (PLS-DA) were performed with metaX (a flexible and comprehensive software for processing metabolomics data) [25]. Partial least squares discrimination analysis (PLS-DA) is a supervised discriminant analysis statistical method. This method uses partial least squares regression [26] to establish the relationship model between the relative quantitative value of metabolites and the sample category to realize the prediction of the sample category. The PLS-DA model of each comparison group was established, and the model evaluation parameters (R^2^Y, Q^2^Y) were obtained through 7-fold cross-validation (seven cycles of cross-validation; when the sample biological repetition number *n* ≤ 3, it was k cycles of cross-validation, k = 2n). R^2^Y and Q^2^Y parameters were used to evaluate the performance of the model, both of which vary between 0 and 1, where 1 represents a perfect fit [27,28]. In order to judge the quality of the model, the model was also sorted and verified to check whether the model is overfitting. The overfitting of the model reflects the accuracy of the model construction, and the overfitting of the model indicates that the model can better describe the sample and can be used as the premise for the model biomarker group search. Overfitting means that the model is not suitable for describing the sample, nor is it suitable for later analysis of this data. The specific method [29] is to randomly mix the grouping marks of each sample before modeling and prediction. Each modeling corresponds to a set of R^2^ and Q^2^ values. Their regression lines can be obtained from the R^2^ and Q^2^ values after 200 times of mixing and modeling. If the R^2^ value is greater than the Q^2^ value and the intercept between the Q^2^ regression line and Y-axis is less than 0, it can indicate that the model is not overfitting, and the model is more stable and reliable [30].

Metabolites with a *p*-value < 0.05 (two-tailed Student’s *t*-test), a VIP > 1, and a fold-change (FC) ≥ 2 or ≤0.5 were considered to be differentially accumulated metabolites. VIP refers to the variable projection importance of the first principal component of the PLS-DA model and represents the contribution of metabolites to the groupings. For clustering heat maps, the data were normalized using z-scores of the intensity areas of differential metabolites and were plotted by the heatmap package in the R language. Z-score (standard score) is a value converted based on the relative quantitative values of metabolites, which is used to measure the relative quantitative values of metabolites on the same level. The Z-score was calculated based on the mean and standard deviation of the reference data set (control group), and the specific formula was expressed as Z = (x − μ) / σ, where x is a specific fraction, μ is the mean, and σ is the standard deviation.

### 2.3. Differentially Expressed Gene (DEG) Analysis

Transcriptome data of leaves infected for 17 days (A) and uninfected leaves by *P. spegazzinii* were downloaded from NCBI (SRR17139378–SRR17139383) and used to explore the differentially expressed genes (DEGs).

To obtain localization information for the reads in the reference genome, clean reads were compared with the reference genome of *M. micrantha* (GCA_009363875.1) [31] using HISAT2-2.0.5 [32], and the expression level was calculated using the fragments per kilobase million (FPKM) method. The differentially expressed genes (DEGs) were analysed using the DEseq2 package version 3.8.6 in R language [33]. Genes with a |log_2_ fold change| > 1 and false discovery rate (FDR) < 0.05 were considered DEGs. The KEGG enrichment analysis of functional significance terms based on the Kyoto Encyclopedia of Genes and Genomes (KEGG, http://www.kegg.jp/kegg/pathway/html (accessed on 17 September 2020)) database was conducted using a hyper-geometric test to find significant KEGG terms in DEGs for comparison with the genome background. GO enrichment analysis of DEGs was performed using the online OmicShare tool (http://www.omicshare.com/tools/index.php/ (accessed on 22 September 2022)). Hypergeometric and FDR multiple tests were utilized to identify significantly enriched pathways. Gene ontology (GO) terms and KEGG pathways with FDR-corrected *Q*-values ≤ 0.05 were considered to be significantly enriched.

### 2.4. Determination of ATP after P. spegazzinii Infection

Following freezing by liquid nitrogen, the samples of infected leaves and uninfected leaves were ground into powder. A subsample of 100 mg powder from each of the three biological samples of infected and uninfected leaves was weighed, and 1 mL buffer solution was added and homogenized in an ice bath using an electric tissue grinder. The mixed extracts were then vortexed and centrifuged at 8000× *g* and 4 ℃ for 10 min. The supernatant was placed into a new Eppendorf tube, and 500 µL trichloromethane was added before being mixed thoroughly and centrifuged at 10,000× *g* and 4 ℃ for 3 min. The upper suspension of each sample was kept and incubated on ice for the subsequent detection of ATP content in infected and uninfected leaves using an ATP content detection kit (Beijing Solarbio Technology Co., Ltd., Beijing, China).

## 3. Results

### 3.1. Metabolite Response of M. micrantha after Infection by P. spegazzinii

A total of 421 metabolites in *M. micrantha* leaves were detected using quasi-targeted metabolomic sequencing, and the diverse set of detected molecules could be roughly grouped into 49 classes, predominantly organic acids and derivatives, amino acids and their derivatives, nucleotides and their derivatives, lipids, phenylpropanoids, and flavones (Appendix A). The PCA showed that the control and treat samples were separated well by PC1, which could explain 39.9% of the total variation (Figure 1A). Partial least squares discriminant analysis (PLS-DA) is a multivariate statistical analysis technique that employs supervised pattern recognition. Its extension of orthogonal projections to latent structures discriminant analysis (OPLS-DA) is a common statistical approach used in metabolomics data analysis. In the PLS-DA model, after 200 permutation tests, the R^2^ intercept of the substitution test in the positive ion mode was 0.97, and the intercept of Q^2^ was −1.56 (Appendix A), suggesting model reliability, given no evidence of overfitting. The PLS-DA scores plot (R^2^Y = 1, and Q^2^Y = 0.7) shows the separation between healthy and infected leaves (Figure 1B).

The significantly different metabolites are listed in Table 1 according to the criteria fold change ≥ 2 or ≤0.5, *p* < 0.05 (*t*-test) and VIP ≥ 1. Metabolites (61 downregulated and 13 up-regulated) significantly changed upon *P. spegazzinii* exposure (Table 1). Among these metabolites, carbohydrates, phenylpropanoids, fatty acids, plant hormones, and the levels of most organic acids and their derivatives, flavonoids, and some amino acids and their derivatives decreased (Table 1). The content of some amino acids and flavonoids increased (Table 1).

The levels of phytoalexin-related metabolites, e.g., maackiain, nobiletin, vasicin, and arachidonic acid, in *M. micrantha* leaves infected with *P. spegazzinii* were 1.73, 2.14, 3.69, and 5.98 times higher, respectively, than those in uninfected leaves (Figure 1C,D; Appendix A). In addition, the level of the stress-related compound, JA-Ile, was 6.40 times higher in infected leaves than in uninfected leaves (Figure 1D; Appendix A).

### 3.2. Differentially Expressed Gene (DEG) Analysis

A summary of RNA-Seq data is shown in Appendix A. This summary reveals that the RNA-Seq data sets are high quality and reliable. After removing adapter reads, ambiguous reads, and low-quality reads, 6.391 Gb clean reads, on average, were produced. High correlations of FPKM value were observed (R > 0.84) between replicates under the same conditions (Figure 2A), indicating that the biological replicates were credible in this study. Principle component analysis (PCA) showed that the gene expression in the *M. micrantha* leaves under two different conditions was clearly separated by PC1 and PC2, which could explain 67.7% of the total variation (Figure 2B). Based on the transcriptome analysis between the *P. spegazzinii*-infected group (A) and non-infected (control) group, a total of 4978 DEGs (15.20% of total genes) were identified, including 2054 significantly up-regulated genes and 2924 significantly down-regulated genes (Figure 2C; Appendix A). These significantly up-regulated genes were significantly enriched in 10 KEGG pathways, including protein processing in the endoplasmic reticulum, ribosome biogenesis in eukaryotes, endocytosis, plant–pathogen interactions, etc. (Figure 2D; Appendix A). The expression of genes related to 20 pathways, including photosynthesis, metabolic pathways, the biosynthesis of secondary metabolites, carbon fixation in photosynthetic organisms, and the circadian rhythm of plants, were significantly suppressed (Appendix A). Further analysis of the up-regulated DEGs showed that these GO terms in biological processes were mainly involved in RNA modification, GO:0009451; macromolecule modification, GO:004341; responses to stimuli (temperature stimulus, GO:0009266; biotic stimuli, GO:0009607; and fungus, GO:0009620); cellular carbohydrate metabolic processes, GO:0044262; jasmonic acid-mediated signaling pathway, GO:0009864; and defense response, GO:0006952 (Figure 2E; Appendix A). Among molecular functions, endonuclease activity, GO:0004519; kinase activity, GO:0016301; phosphotransferase activity, GO:0016773; and hydrolase activity, GO:0016788 were enriched in the up-regulated DEGs (Figure 2E; Appendix A). For the cellular component (CC) category, the up-regulated DEGs in response to *P. spegazzinii* stress showed a clear enrichment in the mitochondrion, GO:0005739, and plasma membrane, GO:0005886 (Figure 2E; Appendix A). The expression of genes associated with photosynthesis, GO:0015979; oxidoreductase activity, GO:0016491; and thylakoid, GO:0009579, were significantly suppressed (Appendix A). All of the GO data are presented in Appendix A.

### 3.3. Plant–Pathogen Interactions

In the plant–pathogen interaction pathway, we found that 63 DEGs were involved in the signal transduction process of the anti-pathogen immune response after *P. spegazzinii* infection (Figure 3; Appendix A). Among them, pattern recognition receptor (PRR) proteins EIX1/2 (EIX receptor1/2), CERK1 (chitin elicitor receptor kinase 1), FLS2 (LRR receptor-like serine/threonine-protein kinase), BAK1 (brassinosteroid insensitive 1-associated receptor kinase 1) and EFR (LRR receptor-like serine/threonine-protein kinase EFR) in the plant-pathogen interaction pathway were significantly induced with higher gene expression levels after rust infection. Calcium-dependent protein kinase (CDPK) and respiratory burst oxidase (RBOH) genes related to ROS production were also significantly up-regulated, which may increase the ROS level in the infected tissues. In addition, expression of the calmodulin (CAM) gene that participates in CAM-dependent signaling pathways was induced to be up-regulated. One mitogen-activated protein kinase 4/5 (MKK4/5) gene was significantly up-regulated 5.3-fold. Similarly, effector-triggered immunity may be triggered by disease resistance protein (RPM1, RPS2), heat shock protein (HSP90), and enhanced disease susceptibility 1 protein (EDS1), which were significantly up-regulated. Studies have shown that they are involved in hypersensitive response, programmed cell death, and defense amplification, respectively.

WRKY transcript factors play significant roles in the regulation of defense responses to pathogen attacks. A total of 17 WRKY genes were significantly up-regulated following infection by *P. spegazzinii* as determined by the RNA-seq analysis. Seventeen DEGs encoded WRKY transcription factors belonging to four different categories (Figure 3; Appendix A). Among them, five WRKY33 and seven WRKY22 members involved in the MAPK signaling pathway were significantly up-regulated by 2.05 times to 15.49 times. In addition, the gene expression of five WRKY52 and two WRKY1 members was significantly induced following *P. spegazzinii* infection, and they may elicit the effector-triggered immune response.

### 3.4. Accumulation of Amino Acid and ATP Content after P. spegazzinii Infection

Metabolome analysis showed that the content of these amino acids, such as L-isoleucine, L-tryptophan and L-citrulline, were significantly increased after *P. spegazzinii* infection (Figure 4A; Appendix A). In addition, the content of most other amino acids, such as L-leucine, histidine, and L-phenylalanine, was also elevated (log2FC > 0), even if they were not significant (Appendix A; Appendix A). Our study showed that the expression of 35 key genes of amino acid synthesis, including branched-chain aminotransferase (ILVE), tryptophan synthase alpha chain (trpA) and acetylornithine deacetylase (AGRE), was significantly up-regulated after *P. spegazzinii* infection (Figure 4A,C; Appendix A).

Interestingly, KEGG and GO enrichment analysis showed that carbohydrate metabolism was significantly induced and photosynthesis was significantly inhibited (Appendix A), while metabolome analysis showed that most of the carbohydrate content was reduced (Table 1; Appendix A). Our study shows that a large number of significantly up-regulated genes function in mitochondria (Figure 2E). Furthermore, the expression of almost all TCA cycle-related genes (8 of 11 key genes) was significantly up-regulated (Figure 4B,C and Appendix A), also causing the amounts of these amino acids to be significantly increased, promoting the production of ATP following infection by *P. spegazzinii* (Figure 4D).

## 4. Discussion

There were both positive and negative responses in terms of levels of metabolites and gene expression when *M. micrantha* was infected with *P. spegazzinii*. While most of the metabolites and gene expressions decreased following infection, there were numerous metabolites and genes that had increased, suggesting that infection by *P. spegazzinii* had stimulated compensatory or defense responses.

Many metabolites detected had significantly higher levels than that seen in uninfected plants. Among all metabolites, the content of amino acids is the most affected, and the content of almost all amino acids is increased, which is a pattern often observed in host–pathogen interactions [34,35,36,37,38]. Amino acids regulate many aspects of plant growth and development as well as biotic and abiotic stress responses. Studies have shown that amino acids play a key role in plant–pathogen interactions and act as precursors for the biosynthesis of defense compounds such as plant antitoxins [39,40]. Amino acid metabolism can also affect the resistance of plant pathogens. For example, the decomposition of the Asp-derived amino acid Lys produces pipecolic acid, which can regulate systemic acquired resistance and mediate the activation of plant defense [39].

The imbalance of amino acids related to the accumulation of homoserine or threonine enhances the immunity of plants to the pathogen of oomycetes [41,42]. In addition, amino acids can also provide resistance for plants through undetermined mechanisms [41]. In the present study, the increased content of most amino acids and four phytoalexin-related metabolites was observed. In addition, amino acids are involved in protein synthesis and can promote and regulate plant growth. In our study, about 40% of the genes identified were up-regulated, and most of these were related to protein processing, suggesting the plant may be trying to maintain growth while infected (Figure 2). The above results show that the increased content of most amino acids may play a role in the resistance and growth maintenance of *M. micrantha.*

Plant defense against pathogens is a process that requires energy [43,44]. ATP generated by the TCA cycle meets the energy demand [45,46]. TCA cycle products also provide a carbon skeleton for the synthesis of amino acids (Figure 4). The amounts of ATP were significantly increased following infection by *P. spegazzinii* infection, as seen in other studies [15] (Figure 4). In addition, the key genes in the TCA cycle presented high expression following *P. spegazzinii* infection. The above results show that *M. micrantha* produces a large amount of ATP to overcome infection and is beneficial to the synthesis of amino acids.

The stress-related compound JA-Ile, which plays an important role in response to the biotic or abiotic stressors [47], was significantly increased by more than 6.4 times following infection by *P. spegazzinii*, again suggesting some sort of defense or compensatory mechanism.

In addition, the key genes in the plant–pathogen interaction signaling pathway presented high expression following *P. spegazzinii* infection. In nature, plants are often attacked by pathogens. However, because the host plant has an effective immune system, the pathogen is perceived by two different recognition systems, which initiate so-called pattern-triggered immunity (PTI) and effect-triggered immunity (ETI), both of which are accompanied by a series of induced defenses [48,49]. PTI is a basal defense response activated by the recognition of PAMPs via pattern recognition receptors (PRRs) at the cell surface. Here, some PRRs were significantly up-regulated under *P. spegazzinii* stress (Figure 3 and Appendix A). Ca^2+^, reactive oxygen species (ROS), MAP kinase cascades, and WRKY transcription factors are considered to be essential components of all PTI-triggered defense responses [48,50]. The increase in cytoplasmic Ca^2+^ is considered to be an important early event in the response signal transduction cascade of plant infection pathogens, and the cyclic nucleotide-gated channel (CNGC) located in the plasma membrane (PM) contributes to the increase in cytoplasmic Ca^2+^during pathogen perception [51]. In *M. micrantha*, four CNGC genes are significantly up-regulated, which indicates that they may mediate Ca^2+^ influx (Figure 3 and Appendix A). CDPKs participate in the phosphorylation and activation of respiratory burst oxidase (RBOH), leading to ROS production [52]. The calcium-dependent protein kinase (CDPK), respiratory burst oxidase (RBOH) genes, and calmodulin (CAM) genes were induced to be up-regulated, which may have activated ROS production (Figure 3 and Appendix A). The MAPK cascade plays a very important role in the signal transduction of plant defenses against pathogen attacks. Arabidopsis AtWRKY22 and AtWRKY29 proteins are essential components of MAPK-mediated plant defense responses against pathogens [53]. In Arabidopsis, WRKY33 plays a key role in the stress response, and WRKY33 overexpression enhances resistance to fungal pathogens [54,55]. In our study, mitogen-activated protein kinase kinase 4/5 (MKK4/5) and WRKY33/22 genes were induced to be up-regulated, which may be related to the activation of plant immunity (Figure 3 and Appendix A). In our study, we also found that RPM1, RPS2, HSP90, and EDS1 were up-regulated dramatically in *M. micrantha* after inoculation. Previous studies have shown that the expression of these genes is associated with the triggering of ETI [56,57]. These results suggested that when *M. micrantha* was infected by *P. spegazzinii*, it activated a defense response, but this does not necessarily imply resistance.

This study has shown that levels of some genes and metabolites increased following infection by *P. spegazzinii*, suggesting that the plant is trying to compensate for or overcome infection. Some of these positive responses relate to resistance, which then raises the question of whether *M. micrantha* could ever become resistant to *P. spegazzinii*, and therefore, will the rust’s impact as a biological control agent lessen?

Despite these positive changes in levels of metabolites and gene expression, this study also showed that *P. spegazzinii* can have a negative impact on gene expression, metabolites, and metabolic pathways. Most of the metabolites (>60%) and genes (60%) had lower levels following infection, suggesting that infection of *P. spegazzinii* has some negative impact on *M. micrantha*.

*Puccinia spegazzinii* is currently being used as an effective biological control agent and has been released against *M. micrantha* in many countries [1,7,11,58,59]. The management of *M. micrantha* using *P. spegazzinii* is seen as a benefit to many landholders, as plants can reshoot after slashing, and herbicides are harmful to humans and the environment [1,11,60]. In both laboratory and field trials, *P. spegazzinii* has been shown to reduce growth rates and flowering of *M. micrantha* [10]. The results in this study at least partially explain what is being observed in the field where growth rates of *M. micrantha* have decreased, offering hope that *P. spegazzinii* will continue to be able to suppress populations of *M. micrantha* in the field.

In this study, we characterized *M. micrantha* as a model host pathosystem for *P. spegazzinii* to investigate the molecular mechanisms of host responses to a pathogen. We have provided the first description genome-wide of the different gene expressions and metabolite levels in the leaves of *M. micrantha* following *P. spegazzinii* infection. Interactions and coevolution between *M. micrantha* and *P. spegazzinii* may influence the overall pathogenicity of rust. Over time, *M. micrantha* may evolve a mechanism to resist *P. spegazzinii*, and the plant could once again become a problem for countries that already have released rust. Alternatively, the rust may develop ways to overcome the plant’s defense responses or resistance. Further studies may provide more insight. The results here certainly provide a better understanding of the interactions between *M. micrantha* and *P. spegazzinii* and the potential of *P. spegazzinii* as a long-term biological control agent for *M. micrantha*.

## 5. Conclusions

In this study, we revealed the response and interaction mechanism of *M. micrantha* to *P. spegazzinii* based on the analysis of plant metabolites and gene expression. The relative levels of essential amino acid, ATP, and phytoalexins were synthesized and accumulated in *M. micrantha* following *P. spegazzinii* infection. In addition, many key genes of the MAPK signalling pathway showed significantly high expression under the *P. spegazzinii* infection.

## Figures and Tables

**Figure 1 microorganisms-11-00678-f001:**
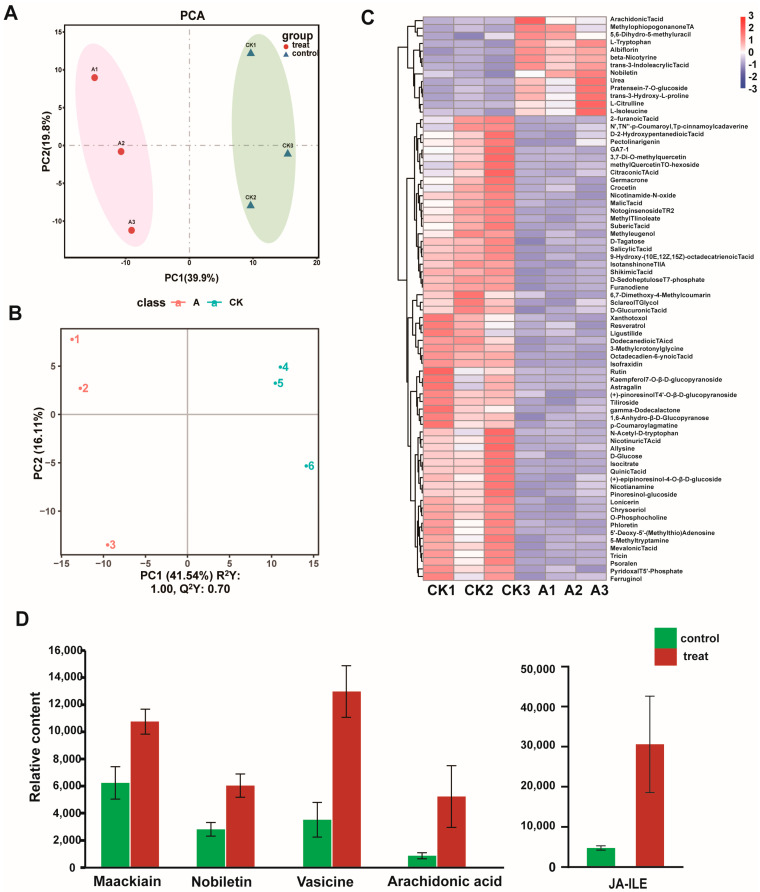
Data analysis of the metabolome in *Mikania micrantha*. (**A**) Principal component analysis (PCA) of metabolome data from healthy and infected leaves of *M. micrantha*. Axes showed the percentage of variance of the first two components (PC1, PC2). The circle represents the treatment group, and the triangle represents the control. (**B**) Partial least-squares discriminant analysis (PL−SDA) plots of untreated (control) and *P. spegazzinii* treated leaves. (**C**) Heatmap hierarchical clustering of differentially expressed metabolites. The content of each metabolite was normalized to complete hierarchical clustering. Each example was visualized in a single column, and each metabolite is represented by a single row. Red indicates high abundance, whereas low relative metabolites are shown in blue. (**D**) Secondary metabolite content associated with immune response in *M. micrantha* under *P. spegazzinii* stress. The error bars represent the means ± SD.

**Figure 2 microorganisms-11-00678-f002:**
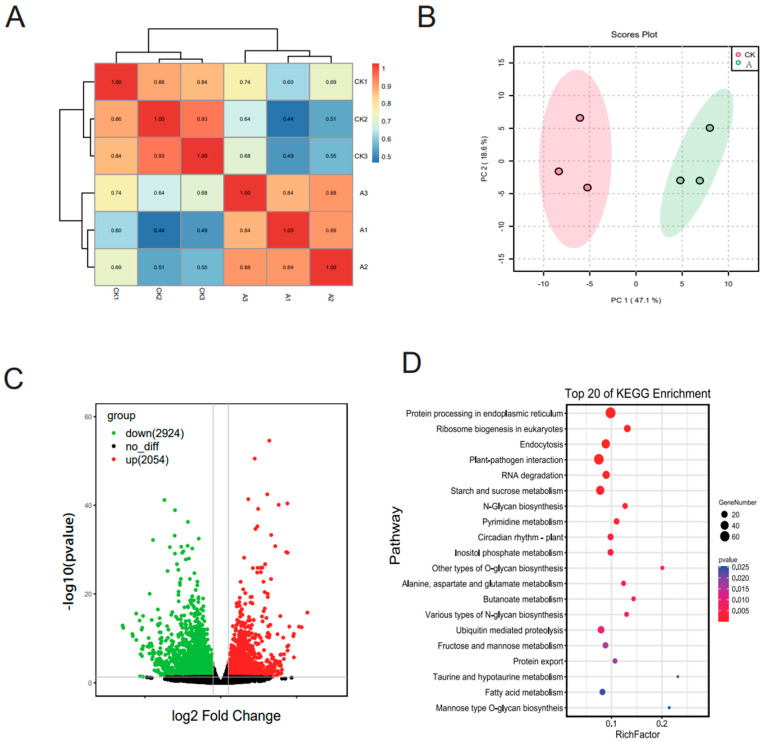
Data analysis of the transcriptome in *Mikania micrantha*. (**A**) Pearson correlations between CK (control) and A (treatment) replicates. (**B**) Volcano map of differential metabolites. Red plots indicate the up-regulated metabolites; Green plots indicate the down-regulated metabolites; Black plots indicate no significant difference. (**C**) Principle component analysis of expressed genes based on the gene expression profiles. (**D**) KEGG enrichment pathway of up-regulated genes. (**E**) GO classification of up-regulated genes.

**Figure 3 microorganisms-11-00678-f003:**
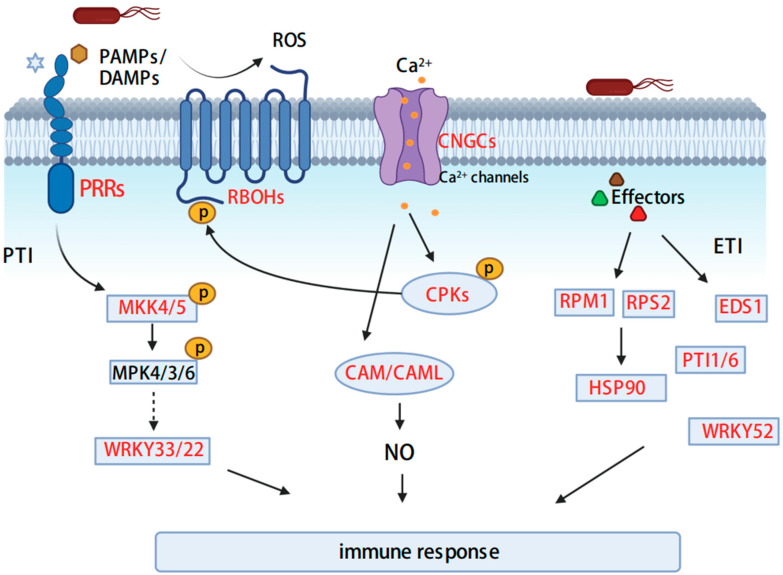
Gene expression of plant–pathogen interaction pathway in *M. micrantha* under infection by *P. spegazzinii.* The red characters indicate that gene expression was significantly up-regulated after *P. spegazzinii* infection. In this figure, we showed how *M. micrantha* responded to the rust infection through the multi-layered specific immune system of plants. The primary response involves the sensing of pathogens through cell surface pattern recognition receptors (PRRs), known as PAMP-triggered immunity (PTI). Upon recognition of PAMPs/DAMPs, downstream components (e.g., RBOHD, CNGCs, MAPKKKs, and WRKY) are subsequently phosphorylated, thereby triggering ROS surge, Ca^2+^ influx, MAPK activation, phytohormone production, and transcriptional re-programming. The second response is called effector-triggered immunity (ETI). Pathogens can directly inject effector proteins into plant cells through the secretion system, and some plants have specific intracellular surveillance proteins (R proteins) to monitor the presence of pathogen virulence proteins. This ETI occurs with localized programmed cell death to stop the growth of pathogens, resulting in species-specific disease resistance.

**Figure 4 microorganisms-11-00678-f004:**
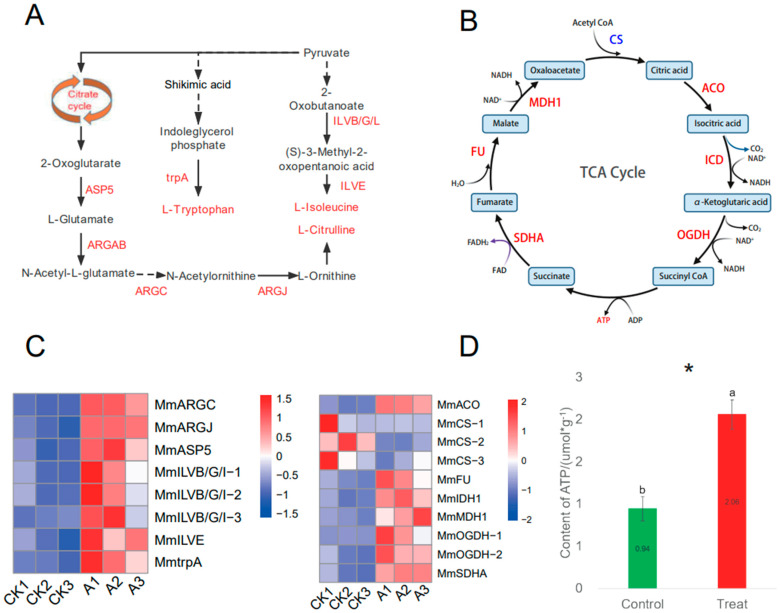
Gene expression and metabolite content of amino acid metabolism and TCA cycle pathway in *M. micrantha* under *P. spegazzinii* stress. (**A**) Pathway of amino acid metabolism; (**B**) pathway of the TCA cycle. The gene expression levels were denoted in blue (down-regulated) and red (up-regulated). Metabolite content is represented in blue (down-regulated) and red (up-regulated). (**C**) Expression pattern of key amino pathway genes in *M. micrantha*; ILVB/G/L: acetolactate synthase I/II/III large subunit; IMS: 2-isopropylmalate synthase; trpA: tryptophan synthase alpha chain; ILVE: branched-chain amino acid; ARGJ: glutamate N-acetyltransferase; ARGC: N-acetyl-gamma-glutamyl-phosphate reductase; ARGAB: amino-acid N-acetyltransferase; ASP5: aspartate aminotransferase, chloroplastic. Expression pattern of key TCA cycle genes in *M. micrantha*; SDHA: succinate dehydrogenase (ubiquinone) flavoprotein subunit; FU: fumarate hydratase, class II; MDH1: malate dehydrogenase 1; CS: citrate synthase; ACO: aconitate hydratase; ICD: isocitrate dehydrogenase; OGDH: 2-oxoglutarate dehydrogenase E1 component. (**D**) Changes in the contents of ATP by the *P. spegazzinii* infection; Significant differences are indicated according to Student’s *t* test: *, *p* < 0.05. The error bars represent the means ± SD.

**Table 1 microorganisms-11-00678-t001:** Significantly altered metabolites in *M. micrantha* leaves under stress.

Type	Downregulation	Up-Regulation	No_Diff	All
All	61	13	349	423
Amino acids and their derivates	4	4	45	53
Nucleotides and their derivates	1	1	36	38
Organic acids and their derivates	9	1	35	45
Carbohydrates	4	0	26	30
Flavonoids	9	3	13	25
Phenylpropanoids	7	0	3	10
Alkaloids	0	1	12	13
Fatty acids	3	1	12	16
Phytohormones	2	0	6	8
Alkaloids	0	1	12	13
Lipids and lipid-like molecules	2	0	11	13
Terpenoids	1	1	10	12
Vitamins	3	0	9	12
Others	16	0	119	135

## Data Availability

The raw sequences data can be found in [NCBI] at [https://www.ncbi.nlm.nih.gov/sra/, accessed on 18 February 2023] under the accession numbers [SRR17139378, SRR17139379, SRR17139380, SRR17139381, SRR17139382 and SRR17139383].

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
