# Peer review of "Metabolomics and Transcriptomics Reveal the Response Mechanisms of Mikania micrantha to Puccinia spegazzinii Infection"

_microorganisms, 2023, doi:10.3390/microorganisms11030678_

Round 1

Reviewer 1 Report

The present ms used metabolomics and transcriptomics to investigate the response of M. micrantha to infection by P. spegazzinii. The results revealed 74 metabolites and 4,978 DEGs. TCA cycle was speculated to be involved in the response. However, the results were too general and non-specific to make any conclusion. The authors claimed that they did the experiment to study the response mechanism of the infection. But the present experimental desgin cannot answer the proposed questions. Overall, the results cannot draw any concrete conclusion. More specific experiment need to be conducted instead of the general analysis that can be done in any plant-pathogen interaction system. A few other concerns are as follows:

L290-291, wrong expression "Moreover, Furthermore, the..." showed unconcious of the writing。

qRT-PCR should be at least conducted to confirm the RNA analysis.

Author Response

  • L290-291, wrong expression "Moreover, Furthermore, the..." showed unconcious of the writing.

Response1: Thanks for the reviewer’s suggestion. We have addressed this to page 12 line 346-347 of the article.

  • qRT-PCR should be at least conducted to confirm the RNA analysis.

Response1: Thank you for your suggestion. However, we believe that the combined research methods of transcriptome and metabolome can adequately support our conclusion, as in the study of [1, 2].

  1. Chen, J.; Le, X. C.; Zhu, L., Metabolomics and transcriptomics reveal defense mechanism of rice (Oryza sativa) grains under stress of 2,2',4,4'-tetrabromodiphenyl ether. Environ Int 2019, 133, (Pt A), 105154.
  2. Barrero-Gil, J.; Huertas, R.; Rambla, J. L.; Granell, A.; Salinas, J., Tomato plants increase their tolerance to low temperature in a chilling acclimation process entailing comprehensive transcriptional and metabolic adjustments. Plant Cell Environ 2016, 39, (10), 2303-18.

Reviewer 2 Report

The manuscript entitled “Metabolomics and transcriptomics reveal the response mechanisms of Mikania micrantha to Puccinia spegazzinii infection” is really  interesting one and the author (s) performed a nice work. I enjoyed the reading of the manuscript. Overall, the manuscript is well structured; presenting novelty and authenticity of work. The results are reliable and manuscript is in accordance with the Journal’s scope. However few grammatical/typo errors are required to fix. Thus, minor revision is compulsory before further consideration.

In abstract, the conclusion of work must be concise and possible future aspects should be illustrated.

All citation need to be numeric in the text do not write the name of authors

In discussion, the author(s) again explained the results (296-303) of this study in some sections that seems redundant as few citations/references in discussion are provided.

The references in general need reformatting according MDPI journal

Author Response

1、In abstract, the conclusion of work must be concise and possible future aspects should be illustrated.

Response: Thanks the reviewer’s suggestion. We revised the conclusions and explained the possible aspects in the future. We summarized the immune response of Mikania micrantha to the stem rust of P. spegazzinii, involving the change of metabolite content and gene expression. Our results can provide a theoretical basis for weakening the defense response of M. micrantha to P. spegazzinii, and for P. spegazzinii as a long-term biological control agent of M. micrantha.

2、All citations need to be numeric in the text do not write the name of authors

Response: Thanks the reviewer’s suggestion. All citation are now rearranged according to MDPI journal requirements.

3、In discussion, the author(s) again explained the results (296-303) of this study in some sections that seems redundant as few citations/references in discussion are provided.

Response: Thanks for the reviewer’s suggestion. We explain the results of the study and provide additional citations/references on page 12 of the article, lines 354-363.

4、The references in general need reformatting according MDPI journal

Response: Thanks for the reviewer’s suggestion. References are now rearranged according to MDPI journal requirements.

Reviewer 3 Report

The focus of the work is an important area - the response of the plant to the pathogen. Mikania micrantha was chosen as the object of study, the significance of this plant as an invasive species is described by the authors in the introduction. The fungus Puccinia spegazzinii was used as a pathogen, which can be used for the biological control of the M. micrantha population. The combination of metabolomics and transcriptomics allows to describe the complex response of a plant complexly and deep. Summing up, the authors reasonably chose interesting and significant objects, aim of study and suitable methods.

However, the text of the article needs improvement.

Description of methods is not sufficient. The LC-MS/MS parameters must be described in detail (equipment, regimen, etc.). Mathematical data processing should be clearly described: normalization, etc. Was P-value adjusted in the metabolomic study? VIPs is mentioned in the results, while there is not a word about it in the Methods part. Is this the result of (O)PLS-DA? Why VIP threshold = 1.74. It is also worth place the description of enrichment analysis in Methods. The parameters of hierarchical clustering (distance, method) should be described. Package was mentioned, is for R language? Authors must specify the software for all methods were used.

In the results a more detailed representation of the data is desirable. It would be well if course of the infection at the time of sampling was be described (phenotypic e . t. c.). Describing the PCA results, the authors presents the cumulative percentage of variance of the first two components, but as can be seen from the score plot, the difference was related only to PC1, so it is only worth talking about the variance of PC1. The authors used VIPs for significant feature selection, but it would be better to give full (O)PLS-DA results. In Table 1, it is desirable to add the total number of metabolites for each class. All bar charts require intervals explanation (SD?). Authors need to specify the units of content. Enrichment analysis for metabolites is could be good. Fig1B is not informative, but if the names of metabolites were added, then coregulated metabolic blocks could be analyzed. The caption for figure 2 is erroneous: there is no description of A and B. It is also advisable to pay special attention to metabolites that are precursors of secondary metabolites involved in the protective response.

The discussion needs to be improved and expanded too. A more detailed comparison of the obtained data with previously published ones is needed. A broader comparative analysis will help to identify the unique and specific features of the response of the studied organism. It would also make the biological sense of the observed changes more understandable. The explanation for the rise in metabolites, especially amino acids, as a way to overcome the effects of infection does not seem convincing completely. For more validity information about growth are needed. In addition, other hypotheses need to be considered. For example, under stress or senescence, protein destruction and the outflow of organic matter and nitrogen from dying organs often occurs. In addition, some amino acids can be substrates for respiration, or routes to gluconeogenesis, or be precursors of secondary compounds. You also need to keep in mind the influence of the fungus itself on the profile of metabolites. Were fungal-specific metabolites recorded in the profiles? Is it possible to estimate the proportion of fungal biomass in the affected organs. In addition, since both metabolomic and transcriptomic studies were combined, it is necessary to compare these results in more detail and speculation about the relationship between changes in the transcript level of enzymes and the level of metabolites, which are their products and substrate could be done.

In general, the work makes a good impression and provides new interesting data on the issues under study. A complex approach allows reveal broad knowledge about the mechanisms of plant response to a pathogen. The resulting model system looks promising. The work may be published after it will be improved.

Author Response

1、Description of methods is not sufficient. The LC-MS/MS parameters must be described in detail (equipment, regimen, etc.). Mathematical data processing should be clearly described: normalization, etc. Was P-value adjusted in the metabolomic study? VIPs is mentioned in the results, while there is not a word about it in the Methods part. Is this the result of (O)PLS-DA? Why VIP threshold = 1.74. It is also worth place the description of enrichment analysis in Methods. The parameters of hierarchical clustering (distance, method) should be described. Package was mentioned, is for R language? Authors must specify the software for all methods were used.

Response: Thanks for the reviewer’s suggestion. The LC-MS/MS parameters have been described on page 3 of the article, lines 109-128. LC-MS/MS analyses were performed using an ExionLC™ AD system (SCIEX) coupled with a QTRAP® 6500+ mass spectrometer (SCIEX) in Novogene Co., Ltd. (Bei-jing, China). The extracts were injected into a column (Xselect HSS T3, 2.5 μm, 2.1 × 150 mm, Waters) with a 20 min linear gradient at a 0.4 ml/min flow rate for the positive/negative polarity mode.

For the mathematical metabolome data, the area of each peak represents the relative content of the corresponding substance, and no special treatment such as normalization was performed. Transcriptome data were generated and processed exactly as described on page 4 of the article, lines 166 to 171. We used FPKM method to normalize gene expression level. FPKM (Fragments Per Kilobase Million), which is the number of Reads from a certain gene per thousand base length per million reads, is a commonly used method of gene expression standardization. This method takes into account the effects of sequencing depth and gene length on gene expression count. This method has been mentioned in line 168-169 of the method section. Other data such as ATP content were not processed.

There were no adjusted P-value for metabolomics studies. Adjust p-value is a more stringent value, which is generally used for omics with large data volume such as gene or transcriptome, and p-value value is generally used for omics with small data volume such as protein or metabolism. We didn't adjust it.

We described the meaning of VIPs on the third page of the Methods section of our article, lines 151 through 152. VIP refers to the variable projection importance of the first principal component of the PLS-DA model and represents the contribution of metabolites to the groupings. I am sorry that our description confused you. When we screened metabolites with significant differences in content, VIP threshold = 1, and we have improved the description to avoid two numbers in two sentences being next to each other, as detailed in line 208 on page 5 of the article. The implementation of (O)PLS-DA and the parameters of hierarchical clustering have been described in detail in lines 145-160.

2、In the results a more detailed representation of the data is desirable. It would be well if course of the infection at the time of sampling was be described (phenotypic e . t. c.). Describing the PCA results, the authors presents the cumulative percentage of variance of the first two components, but as can be seen from the score plot, the difference was related only to PC1, so it is only worth talking about the variance of PC1. The authors used VIPs for significant feature selection, but it would be better to give full (O)PLS-DA results. In Table 1, it is desirable to add the total number of metabolites for each class. All bar charts require intervals explanation (SD?). Authors need to specify the units of content. Enrichment analysis for metabolites is could be good. Fig1B is not informative, but if the names of metabolites were added, then coregulated metabolic blocks could be analyzed. The caption for figure 2 is erroneous: there is no description of A and B. It is also advisable to pay special attention to metabolites that are precursors of secondary metabolites involved in the protective response.

Response: Our metabolome content and gene expression are relative, so it is appropriate and sufficient for us to describe the changes by fold in our results.

Thanks for the reviewer’s suggestion. We described the course of infection at the time of sampling on page 3 of the article, lines 98 to 102. In the new version, we only describe the variance of PC1 and give the complete (O) PLS-DA results. In Table 1, we added the total number of metabolites in each class. The error bars represent the means ± SD.

We used the Quasi-Targeted metabolomics technology to determine the metabolite content, which can only obtain the relative content and cannot specify the unit of each metabolite content. We carried out kegg enrichment analysis on metabolites with significant changes in content, because there are few kinds of metabolites with significant changes in content, we have no way to obtain significant enrichment. In Figure 1B, we added the name of the metabolite, but we did not find valuable information in the coregulated metabolic blocks.

Thanks for the reviewer’s suggestion. We have re-described the caption of Figure 2. See lines 217-226 on page 6 for details.

3、The discussion needs to be improved and expanded too. A more detailed comparison of the obtained data with previously published ones is needed. A broader comparative analysis will help to identify the unique and specific features of the response of the studied organism. It would also make the biological sense of the observed changes more understandable. The explanation for the rise in metabolites, especially amino acids, as a way to overcome the effects of infection does not seem convincing completely. For more validity information about growth are needed. In addition, other hypotheses need to be considered. For example, under stress or senescence, protein destruction and the outflow of organic matter and nitrogen from dying organs often occurs. In addition, some amino acids can be substrates for respiration, or routes to gluconeogenesis, or be precursors of secondary compounds. You also need to keep in mind the influence of the fungus itself on the profile of metabolites. Were fungal-specific metabolites recorded in the profiles? Is it possible to estimate the proportion of fungal biomass in the affected organs. In addition, since both metabolomic and transcriptomic studies were combined, it is necessary to compare these results in more detail and speculation about the relationship between changes in the transcript level of enzymes and the level of metabolites, which are their products and substrate could be done.

Response: Thanks for the reviewer’s suggestion. We have improved the Discussion and compared the data obtained with the data released before in more detail, especially the increase of amino acids.

As you said, whether the increase of amino acid content is due to the destruction of protein and the outflow of organic matter and nitrogen from dead organs cannot be determined, but this is a very worthy research direction.

In addition, some amino acids can be the substrate of respiration, the pathway of gluconeogenesis, or the precursor of secondary compounds. In our study, through previous studies, we believe that amino acids may contribute to the synthesis of plant antitoxins.

Puccinia spegazzinii can't survive without its host, so we can't record the fungus-specific metabolites.

We are unable to estimate the proportion of fungal biomass in affected organs.

Round 2

Reviewer 3 Report

It is worth noting that the authors did a sufficient work to improve the manuscript and now it gives an impression of more a completeness. I am satisfied with the responses of the authors on my comments.

Just few notes:

firstly, captions of figure 2 still is not complete (first correlation heatmap is nod described)

secondly, Q2>0.5 is more appropriate threshold for PLS-DA, maybe more relevant (than 26) reference for PLS should be made.

The manuscript may be published after minor tweaks. I wish the authors good luck in their studies.

Author Response

1、Firstly, captions of figure 2 still is not complete (first correlation heatmap is nod described).

Response1: Thanks for the reviewer’s suggestion. We have modified this to page 6 line 235-236 of the revised version.

2、Secondly, Q2>0.5 is more appropriate threshold for PLS-DA, maybe more relevant (than 26) reference for PLS should be made.

Response2: Thanks for the reviewer’s suggestion. Partial Least Squares Discrimination Analysis (PLS-DA) is a statistical method of supervised discriminant analysis. The method uses partial least squares regression to establish a relationship model between the relative quantitative values of metabolites and sample categories to achieve the prediction of sample categories. A PLS-DA model was established for each comparison group, which was validated by 7-fold cross-validation (seven times cross-validation; When the number of biological replicates of the sample is n≤ 3, it is k cycles of cross-validation, k=2n) to obtain the model evaluation parameters (R2Y, Q2Y). If the values of R2Y and Q2Y are closer to 1, the model is more stable and reliable. As you said, Q2>0.5 is more appropriate threshold for PLS-DA. What you said Q2 corresponds to our Q2Y (Figure 2). Value of Q2Y is greater than 0.5, indicating that our model is stable and reliable.

In order to distinguish the quality of the model, we also rank and verify the model to check whether the model is "over-fitting". Whether the model was "over-fitting" reflected the accuracy of the model construction, while no "over-fitting" indicated that the model could describe the sample well and could be used as a prerequisite for finding biomarker groups in the model. "Overfitting" means that the model is not suitable to describe the sample and is not suitable for subsequent analysis of the data. The specific method is to randomly shuffle the grouping marks of each sample and then conduct modeling and prediction. Each modeling corresponds to a set of values of R2 and Q2, and their regression lines can be obtained according to the values of R2 and Q2 after 200 times of shuffling and modeling. When R2 value is greater than Q2 value and the intercept between Q2 regression line and Y-axis is less than 0, You can indicate that the model is not "overfitting", as shown in Supplementary Figure 2.

According to your suggestion, we have made more references. See lines 147 to 167 on page 3-4 of the revised version.
